# The Attitudes of Teachers towards Disadvantaged Young Students: Israel–Romania Comparative Analysis

**Pazit Levi-Sudai [1] and Gabriela Neagu [2,*]**

1  Faculty of Social Sciences and Humanities, Oranim Academic College of Education, Tivon 3600600, Israel
2  Research Institute for Quality of Life, 050711 Bucharest, Romania
*  Correspondence: gabi.neagu@iccv.ro; Tel.: +40-72-341-763-9

**Abstract:** The aim of this article is to identify and analyze the attitudes of teachers toward disadvantaged young students (DYS) in two different social, economic, and cultural contexts: Israel and Romania. In terms of methodology, we used a qualitative method—focus group—for data collection. Focus groups were organized around open-ended questions that were designed to encourage teachers to reflect on the situation of DYS and their attitudes towards them. From a theoretical point of view, our analysis began with the description of the meanings of four central concepts: attitude, educational integration, inclusive education, and DYS. The results of our research confirmed the fact that knowledge, feelings, and behaviors are widely recognized as the three pillars of teachers' attitudes towards DYS. Through the comparative analysis of two different educational systems, we highlighted that teachers' attitudes towards DYS can be influenced by structural elements—level of socioeconomic development, and historical and cultural specificity—but also by individual elements: the training of teachers to work in different educational contexts, the level of support received, and the type of school. The results of this research can represent a source of relevant information for the policies and practices of quality education for all.

**Keywords:** attitude; disadvantaged young students; education system; Israel; Romania

## 1. Introduction

Education for all is an objective and principle that is supported by international organizations (UN, UNESCO) since the early 1950s. It has been continuously improved so that all children, regardless of their particularities, can benefit from a quality education. In this sense, but also to ensure access to education for all categories of young people, quality education for all is one of the priority objectives of the 2030 Agenda for Sustainable Development Goals (SDGs) that was signed in 2015 by the United Nations member states [1]. Of the 17 sustainable development goals, one (SDG 4, "Quality Education", in particular, point 4.5: "By 2030, eliminating gender disparities in education and ensuring equal access to all levels of education and training for people vulnerable people, including people with disabilities, indigenous peoples and children in vulnerable situations") aims to change attitudes towards DYS so that they can benefit from quality education. All of these recommendations are necessary because it is a proven fact since the early 1960s that students who belong to disadvantaged socioeconomic, familial, cultural, ethnic, etc., environments, have a lower chance of accessing and succeeding in education [2–4]. In different studies [5], we found the idea that teachers tend to have less favorable attitudes towards DYS and lower educational expectations, and consider them to be less academically oriented and more exposed to the risk of dropping out of school. The assumption is that socioeconomic factors, family context or the eventual psychophysical and intellectual disabilities do not help DYS to face scholastic requirements. Other researchers [6] mention that most studies focus on the relationship between students' socioeconomic status and scholastic results, and in most cases, the conclusion is that external factors are responsible

for student success or failure in school (employment and education of the parents, family size, belonging to an ethnic minority). "School makes a difference," [7,8] is the idea that the impact of a student's background, or certain psychophysical and intellectual characteristics on the student's scholastic development, can be limited by improving learning conditions; this includes changing teachers' attitudes. "School makes a difference," is supported by teachers' confidence in their skills and knowledge that they can influence how well students learn, even those students who are considered difficult, who have special educational needs (SEN), who are disadvantaged socioeconomically and culturally, and who have difficult family backgrounds.

The aim of our article was to investigate the attitudes of teachers towards DYS and it is submitted to the following research question: what are the factors and the conditions that influence the attitudes of teachers toward DYS? The proposed analysis was a comparative one between two countries, and two education systems—Israel and Romania—and was based on qualitative research. Through comparative analysis, we focused attention on the implications of differences and similarities that could lead to improvements in the cross-cultural dialogue and improvements in understanding teachers' attitudes towards DYS. Furthermore, through this type of analysis, we had the opportunity to analyze our own beliefs, convictions, and behaviors toward DYS in national contexts, and to critically reflect upon our own attitudes towards this issue.

The article is structured into the following sections: contextualization, in order to better understand the issue of attitudes toward DYS, a short description of the specifics of each education system in the two countries was included in the analysis (2); part (3) and part (4) are allocated to the description of the theoretical and methodological framework underlying the analysis, with an emphasis on the definition of concepts, and on an explanation of the methodology used; (5) the results obtained are presented and analyzed; in (6) we discuss the limits of the research; and in (7) are conclusions and implications for practice.

## 2. Contextualization

In general, there was a tendency for cross-national studies to focus on a small number of with similar situations of socio-economic development, with educational systems assumed to be performing well or having appropriate or identical models of organization. Therefore, researchers often compare the education systems of countries in northern Europe—Norway, Finland, Denmark, Sweden [9,10], Germany, Switzerland, and Austria [11,12], or in former communist countries [13]. Other studies consider non-European countries with education systems that are close in performance, making comparisons either between the United States and Israel, for example [14]. We believe that the development of studies that include countries that differ in terms of the organization of the education system, the level of socio-economic development, and culture, would facilitate awareness and importance of the diversity of national contexts and their potential implications for the school population and for teachers. Also, such studies are capable of contributing to the identification of solutions to common problems that manifest themselves in different educational contexts.

Israel and Romania are two different countries, in terms of the organization of the education system, but also differ from a socioeconomic and cultural point of view. At the same time, they also have common elements: public education at all levels, for example. The description of the educational and sociocultural contexts in which the teachers from the two countries carry out their activities is very important to understand their attitudes towards DYS, especially since for specialists [15] the attitude is a social construct. Moreover, some researchers [16] are of the opinion that studies on teachers' attitudes towards disadvantaged students, whatever their disadvantages may be, have not fully considered the impact that cultural and social-historical characteristics can have on this process. Our research complements this perspective.

There are several characteristics of the State of Israel that need to be considered, since these can shed some light on teachers' perceptions and attitudes. Israel is a young country

that has absorbed immigrants from many countries (and still does) since its inception in 1948. Therefore, its population is characterized by ethnic diversity, immigrants and veterans, and citizens of different religions. The complexity of Israeli society and its diverse human tapestry are reflected in its education system. As a result, there is great diversity among schools located in various parts of the country, as well as within classrooms [17]. It is customary to present the structure of the Israeli education system as being divided into four main sections: age (education stages), the legal status of the educational institution, type of inspection, and sector (Muslim, Christian, Druze, Jewish—state, state religious, ultra-Orthodox). Resulting from this branched system is a diverse range of schools and educational frameworks within and outside communities: boarding schools, youth villages, supervision framework in the Ministry of Welfare and Social Affairs, and the Ministry of Economy and Industry, alongside those of the Ministry of Education. The education system's power and ability to perform derive prominently from legislation regarding the outline of the education system in the country.

Romania has had a different path in terms of the process of social and educational integration, one that has been largely marked by socioeconomic and political evolution. The establishment of the totalitarian regime in the second half of the 1940s also meant an orientation toward the homogenization of the population, even if the people were very different, with different situations and different social, economic, racial, religious, and ethnic origins. In communist Romania, the acceptance of people with characteristics different from the majority or of poverty was associated with a failure of the regime, leading to a situation in which these categories of the population had to made "invisible": isolating them in special institutions (schools, orphanages, hospitals, etc.), and not recognizing their language, ethnicity, different cultures or problems related to psychophysical and intellectual conditions. The Romanian society underwent an extensive reform process immediately after the change in political regime in 1989. However, the Romanian educational system is rather elitist instead of inclusive, because it focuses more on competitiveness and performance [18], and less or not at all on students' needs or on equality and equity in education. Punishment or appreciation of teachers in educational institutions is based on the performance of students in various competitions, while the success of DYS in the process of integration or inclusion is not considered. Even if certain reforms led to significant changes in education and in the organization and functioning of the system, they rarely, or not at all, influenced attitudes of those who worked in schools in Romania. Similarly with other states from the former communist bloc, in the Romanian education system, "the practice is still lagging behind the legislation." [19] (p. 1266).

## 3. Theoretical Framework

The following review addresses the basic concepts of this research. The concept around which we built our research is that of attitude. The next three pillars of research are inclusive education, educational integration, and disadvantaged young students (DYS).

### 3.1. Attitude—Definition, Components

Although attitude is a concept that is often used both in specialized works and in everyday discourse, there is no unanimously accepted definition among specialists; there are only common views on what attitude means. Attitude is defined as a general and lasting feeling, positive or negative, towards certain people, objects, or situations [20]. This type of definition emphasizes the duration of attitudes and their persistence at the individual and societal levels. Other definitions [15] highlight the social character of attitudes: individuals act or react according to their beliefs, values, or paradigms that they have acquired through their social experiences.

One of the most common definitions is that of Eagly and Chaiken [21] according to which attitude is a psychological tendency, expressed by a favorable or unfavorable assessment of a situation, object, or class/group of situations, or objects. The two researchers emphasize the evaluative factor of attitude. Evaluation can be expressed cognitively (ideas

and opinions about situations/objects/people), affectively (emotions and feelings about the attitudinal object), or it can be expressed behaviorally (intentions of action, active behavior towards the attitudinal object). Applying Eagly and Chaiken's definition in this article would translate as follows: the vast majority of teachers have beliefs, and have a set of knowledge about DYS (cognitive component), but these may reflect indifference or disinterest in this category of the population; or contrarily, teachers may have a range of feelings and emotions—fear, rejection, compassion, acceptance (affective component). The behavioral component is manifested when teachers carry out certain actions for or against DYS: Teachers learn or work with DYS, spend free time with them and keep in touch with these students and their families, or choose to carry out their activities in spaces and institutions that do not are frequented by DYS, or are difficult to access for this category of students. In other words, if teachers have the belief that all students, regardless of their characteristics, have the right to education, then their attitude towards educational integration and educational inclusion will be favorable. If teachers assume that it is their responsibility, for each student, regardless of his/her socioeconomic, psycho-intellectual, ethnic, or religious characteristics, to become successful in education, then teachers' attitudes will be favorable. On the contrary, the belief that students are responsible for their own success or failure will lead to a negative attitude of teachers towards students that do not live up to expectations.

### 3.2. Disadvantaged Young Students

Disadvantaged young students (DYS) are the category of the population whose balance of social, family, economic, educational, and psychological factors is affected by the occurrence of a difficult situation that they cannot solve with their own resources. This lack of resources (material-financial, psycho-intellectual, etc.) is most often identified as the main cause of difficulties DYS have with educational integration or educational inclusion. The literature includes numerous studies that demonstrate significant associations between the socioeconomic situation of the family and the educational success of the children [22,23]. DYS not only lack the resources to purchase books or school uniforms, but they also have a narrow language code that is incompatible with the elaborate language code used in schools [24], making it difficult to integrate into school culture [3]; this affects their relationships with peers and teachers. The psychosocial development of young people is also strongly influenced by the socioeconomic conditions in which they live [25], suggesting that DYS have a higher risk of showing behavioral and emotional problems such as rebellion and impulsivity, and are more difficult for colleagues and teachers to understand. Also, when we talk about DYS, we have to take into account that students with special educational needs (SEN) are also included in this category. Students with SEN are more vulnerable to exclusion compared to disadvantaged socio-economic, family, cultural students, even for the simple fact that they are perceived as a dependent, and less productive and competitive in an organizational context (school, workplace, community) [26]. For DYS, the educational environment becomes in many situations much more important than the family environment: coming from disadvantaged environments, these students do not see their parents as a source of support. Several studies [27,28] have shown the major importance of classmates, especially when it comes to the educational success of DYS. Classmates and school are a source of social and emotional support, especially when such sources give the disadvantaged opportunities to identify with and belong to a group that is often superior to the one they come from [29]. In order to succeed in teaching, the staff who work with DYS must not only have scientific skills, but also skills of a relational and socio-emotional nature. A collection of research [30] has come to the conclusion that for DYS, the affective dimension of the instructional–educational process is much more important in schools than in schools where mostly favored students learn from socioeconomic, family, cultural views, etc.; the reason is because the former category of students is more sensitive to encouragement, praise, and emotional support compared to the latter category.

In conclusion, in order to succeed in didactic activities, teachers who work with DYS must not only have scientific competencies, but also relational, socio-affective, etc., competencies.

### 3.3. Educational Integration and Educational Inclusion

Educational integration and inclusive education are two different concepts. Some researchers believe that integration most often means a process of assimilating students who are in difficulty [31], or placing students in a pre-existing structure to which they must adapt. School inclusion is defined as a situation in which all school-aged children, regardless of their psycho-intellectual, psycho-physical, socio-familial, economic factors, etc., attend the classes of a regular school, and actively participate in the social and educational life of the group of students to which they belong [15,32,33]. Other researchers [34] argue that inclusive education must relate to social justice, equity, and equality in society, while still others [35] argue that inclusion means the same thing as quality education for all. Despite the necessity for definitions, the fear of oversimplifying the concepts is clear [36]; therefore, consideration should be given to the need for, and the importance of, examining teachers' attitudes toward integration and inclusion, and thereby deepening our knowledge and understanding about how teachers view the perceptual and practical meaning of including at-risk vulnerable students in schools. Furthermore, those studies that talk about the inclusive education mostly consider students with SEN (usually of a psycho-intellectual, psycho-physical nature) and, to a lesser extent, the socioeconomic situation. Razer and Friedman [37] (2017) defined excluded students to be those who have the cognitive ability to succeed but are trapped in a cycle of failure that makes their educational work difficult. The use of the concept of exclusion describes the reality of children who are characterized by failure, disruptive behavior, and isolation [37]. Schools often operate, whether intentionally or not, as agents of social exclusion rather than as agents of social inclusion; hence, DYS and teachers experience ongoing failure, and are trapped in relationships of mutual rejection and isolation. Schools that have students from backgrounds of poverty, suffering or social marginalization sometimes exhibit problems such as learning avoidance, contact avoidance, defiance, violence, overstepping and the like. The inclusive educational approach refers to these behaviors as an inherent part of the risk of exclusion but which can be avoided through specific measures including changing the attitude of teachers [37].

Although it is obvious that both educational integration and especially inclusive education support the SDGs for DYS [1], ensuring access to quality education for all categories of students, regardless of their characteristics, depends on the attitudes of education systems, especially those of teachers.

## 4. Materials and Methods

The present research is part of a project financed by the Faculty of Social Sciences and Humanities, Oranim Academic College of Education, developed between 2021 and 2022. Our intention in this project was to triangulate the perspectives of different key actors (teachers, other categories of specialists working with schools, and future teachers) regarding the scholastic integration and inclusion of DYS, in order to identify consistencies and inconsistencies of opinions, and ultimately establish the impact of these actors' attitudes on the process of integration and school inclusion.

### 4.1. Methods

Data were collected on the basis of the focus group (FG) methode and using subsequent content analysis. We opted for a qualitative method because it facilitates the gathering of information to characterize teachers' attitudes (*what they feel*), their knowledge (*what they know*), and their activities (*what they do*) [38] (p. 330). Experts recommend the FG methode to study various attitudinal topics, such as teachers' beliefs about school inclusion [39] and about overcoming historical, cultural, socioeconomic, and political barriers when working with DYS [40]. Compared to other qualitative data collection methodes techniques (for example, semi-structured direct interview or observation), the FG is a suitable method

because it offers the perfect framework for respondents to describe their experiences, practices, and strategies which they adopt in classes and schools in a particular activity, in direct relation with DYS, and to describe the arguments that were the basis of their choices, decisions, and behaviors.

The research carried out in the project was extensive, and included the development of FGs with students (to understand how future specialists relate to DYS) and teachers who already apply what they learned during their training and from their professional experience. We also included the target groups and other categories of specialists who are connected to the issues of DYS (social workers, counselors, psychologists). In the next stage, researchers from the two countries developed three categories of tools: an FG guide for teachers, an FG guide for other types of specialists, and an FG guide for students. Given the complexity of the research topic, we selected and analyzed only the data collected from the FGs that involved the teachers from both countries.

In building the FG guides, we took into account each attitudinal component in formulating questions for the teachers from the two countries. The cognitive component was based on confidence in professional skills and educational background, and we generated questions such as the following: *Are you prepared to work with DYS? Do you think that you need special skills, and competencies to work with DYS?* For the affective component, we emphasized the comfort level of specialists in working with DYS, and posed questions such as the following: *Do you feel comfortable working with DYS?* How would you characterize the relationship between teachers and DYS? For the behavioral component, we referred to teaching methods, collaborations among teachers and with DYS and other specialists, or collaboration with DYS families; some resulting questions were the following: *Do you use different teaching-learning methods for this category of students? Do you collaborate with the families of DYS? Do you ask for the help of your colleagues?* We video recorded and transcribed all of the FGs, and then verified all of the transcriptions. The content of the FGs was transcribed in Romanian and Hebrew, and then translated into English. Each researcher verified the transcripts collected from the two countries. The next stage was the development of a system of analysis categories that were composed of themes and sub-themes which we used to classify the sentences from the transcripts. In creating the system of themes and sub-themes, we took into account the defined components of the attitude: cognitive, affective, and behavioral. Based on a frequency analysis, we ordered the themes and sub-themes to be analyzed. Themes were identified on the basis of the frequencies with which they were mentioned in the discussions with the research participants, with sub-themes identified as those that were associated with the central themes of the research. In order to carry out the content analysis, we considered the following rules: compliance (the data were collected using identical methodology in both countries, the same content of the FG guide, the same meaning of the concepts), mutual exclusion (each sentence was included under a single theme and sub-theme), relevance (the analyzed data concerned only the subject studied—the attitude towards DYS), and objectivity (the sentences reflected the definition of each theme and sub-theme) [41]. In the case of themes and sub-themes that registered a lower frequency, we re-analyzed them to see if they did not reflect different nuances of the initial themes and sub-themes. Classification, review, and re-evaluation of all themes and sub-themes were carried out independently by the researchers from the two countries: initially, each researcher analyzed the content of the FG from his country so that later, through the exchange of data, each one carried out the same analysis for the collected data by the other team.

*4.2. Sample Selection and Participants*

In order to select the participants, in the first stage, we sent an email to some teachers that contained information about the project (data about the people responsible for the research, the purpose of the research, the method of data collection, the duration of the FG, characteristics of the target group, the time available to confirm participation in the FG). We encouraged the teachers to send this email to their colleagues and collaborators, taking into

account the selection criteria. In the next stage, we created a list of teachers who expressed their willingness to participate in the FG, in order to analyze their socio-demographic and professional characteristics. In the final stage, we selected participants for the FG, and sent an email with information about the date and time when the FG would be held. The research included the participation of 10 teachers from Romania, and 16 teachers from Israel, with different socio-demographic and professional characteristics. The education system in Israel is much more complex in terms of ethnic and religious minorities, compared to the education system in Romania. In order to respect the multidimensional, multicultural character of the Israeli education system, a larger number of FG participants was needed. The increase in the number of participants in the FG in Israel also supported the cross-cultural perspective through which we analyzed the research data.

Participants in the FG in Israel were regular high school, vocational high school, SEN school, SEN classroom, middle school, elementary school, high school yeshiva (where religious boys study), and anthroposophical school teachers. In terms of the level of education and the disciplines, the subjects taught in regular classes, in special education classes, in classes for students with learning/emotional and behavioral difficulties that are not defined as special education; they taught math, language, and English. Another characteristic was seniority in educational work—early stage teachers to teachers with rich experience in education (1–23 years of experience). Their age range was 27–63 years. In Israel, most of those who are involved in teaching are women; this characteristic is also reflected among the participants, who comprised 5 men and 11 women. Meanwhile, the FG in Romania was organized by teachers, and attended by 10 people. In order to obtain a more realistic picture of the types of attitudes of the teachers towards DYS in the FG, we included representatives from all levels of education: compulsory education (primary and secondary), secondary education, and university education. We also considered the participation of teachers from mainstream schools as well as from inclusive schools (educational institutions that are attended by students with special needs), teachers who work in educational institutions in rural areas, and teachers from urban areas. Teachers participating in the FG came from schools that are perceived as prestigious schools—from the perspective of school results, as well as socioeconomic and family characteristics of the school population—but also from schools that operate in isolated communities (those in rural areas). We also considered participants from schools with diverse populations consisting of students from disadvantaged backgrounds, disadvantaged students, students with special needs, those belonging to ethnic minorities, etc. From the point of view of professional experience, the teachers had at least 5 years of experience in teaching to over 25 years. The demographic characteristics of the FG in Romania reflected the situation at the national level: the majority of teachers were women (9 women and 1 man), and the age range was 25–55 years.

*4.3. Data Collection*

In establishing the research protocol and data collection, the researchers [42] suggests taking into consideration some circumstances or factors that ensure the trustworthiness and credibility of the research: personal, interactional and contextual factors. The first category of factors—personal—included the moderators and teachers participating in the FGs. In the case of the moderators, the authors of this article moderated all of the FGs. For this research, both moderators used personal connections to invite the participants in the FGs and there is a risk that the teachers' answers will be influenced by this aspect: having close relationships, people tend to know their opinions on certain issues and may have a tendency to formulate their answers depending on what they think the ohter expects. The experience of moderators in collaboration with the teachers and DYS, the experiences in FG moderation, and the in-depth knowledge of the research project of which they are the authors are in a position to reduce significantly these risks. Regarding the teachers, the main factor that had the potential to influence the research data was the selection method: the snowball methode. The main advantage is the easy access to the target group, with the

trust between the moderator and the target group, but also within the target group, because it was based on recommendations from well-known people. Regarding the disadvantages, this selection method facilitated the inclusion in the sample of those teachers with interrelationships, those who are more involved in the school and in the educational life of the institution, and those who participated in various professional courses; on the other hand, the selection method "loses" those who were "isolated" (teachers on the verge of retirement, those who worked in schools in isolated localities with no access to ICT, or did not have digital skills). In order to limit the disadvantages of the selection method, we considered expanding the selection base of the teachers in order to have access to as large a number as possible, so that through the characteristics of the participants, we could reflect the characteristics of the education systems in terms of gender distribution, education levels, types of schools, age, and professional experience.

As for the interactional factors, a challenge came from the fact that each participant tended to bring others with whom they worked and collaborated with; thus, it is very possible to have common opinions and attitudes, which could lead to the creation of minorities or alliances between participants during FGs; these interactional factors could also generate processes of social conformity among the other participants [21]. In order to control the influence of familiarity, we invited teachers from different regions and cities, as well as different schools, to ensure that a balance was achieved between the similarities and differences of the characteristics of the participants in each FG.

Regarding the contextual factors, we considered the place or context in which we conducted the FGs: online meeting. The Google Meet and ZOOM platform offers better facilities for carrying out FGs compared to only audio recordings, because it also includes images as well as the possibility for participants to see one another; furthermore, the chat function of these applications offers the possibility of making comments, synchronously or asynchronously, without interrupting the discussion or discourse of participants. Such comments, suggestions, and questions are visible to all FG participants in real time. Moreover, this facility is useful for the moderator, who can repeat or develop an idea suggested by one of the FG participants. At the same time, however, we must take into account the fact that not all people feel comfortable from a psychological point of view with these online platforms; the fact that each participant was in a different physical environment (at home, at the office) created the conditions for disruptive factors (noises, extraneous discussions with colleagues or family members). In order to compensate for this aspect, we extended the duration of the FGs to offer opportunities for each participant to express their opinions, or to have the necessary time to manage any disruptive factors. We also presented a minimum of rules to mitigate disruptive conditions: using the chat for comments, opening the microphone only when there was going to be an intervention, closing the microphone and the video camera if there was a discussion or event that did not take place within the scope of the research activity, announcing interventions by using other facilities of the platform, raising a hand.

We organized the FGs according to three stages: introduction, discussion, and conclusion. In the first part of the FG (introduction, around 15 min) we exchanged general opinions to create an atmosphere that was suitable for this type of discussion (relaxed, open). After connecting all the participants, each of them introduced themselves, including the moderator. At this stage, the details of the research project were also shared, its objectives, and the fact that the data collection was being carried out simultaneously in another country. Also, the rules of organization were presented again, and each participant expressed both verbally and in writing (via chat) their agreement to be audio-video recorded. The FG participants were also informed that there were no positive/good answers and negative/wrong answers, only the opinions based on their professional and personal experience. All opinions were important for the discussion. In the second part— the discussion—the moderator asked a question or launched an idea to the debate, and ensured that each participant in the FG provided an answer, and expressed an opinion regarding the question or idea. Although the discussions were guided by open-ended

questions that were organized around pre-established themes, personal experiences and spontaneous issues raised by the research participants also served as starting points for exploring the central theme of the research. In concluding, the moderator delivered a summary, and asked the participants to provide feedback so that possible biases of the moderator could be avoided, and so that each group could validate the main conclusions.

### 4.4. Ethical Consideration

At this stage, the researchers who were involved in the project conducted meetings in which they discussed the content of the questions. They endeavoured to maintain consistent meanings and significances in both the Hebrew language and in the Romanian language, and to have the same meaning for the population that participated in the research. The next stage consisted of the research instruments that were submitted for the analysis, and for approval by the Ethics Committee of the faculty. Each FG lasted between 1 and 2 h. The participants were informed that the discussions would be recorded, and that the data would be used exclusively for scientific purposes. In the data collection activity, the rules regarding the protection of personal data and the anonymization of the answers were respected. We established a timetable for data collection, simultaneously in both countries (February–March 2021).

## 5. Results and Discussion

The research data allowed us to analyze the attitudes of teachers towards DYS on the main components—affective, cognitive, and behavioral—as well as make comparisons between the two countries. Statements from the FGs and their organization into three major themes (attitudinal components) and different sub-themes, helped us to understand the particularities and needs of the two education systems in the matter of DYS.

### 5.1. The Cognitive Component

The attitudes of teachers towards DYS, from the perspective of the cognitive component, is conditioned by the level of information and knowledge that teachers have towards this category of students; they are affected by the confidence they have in their abilities to teach and interact with the students, their training in this area, etc. Some researchers [43,44] believe that favorable or unfavorable attitudes toward or against an object, situation, or event, are the result of the information that people hold about each other. In order to analyze the cognitive component of attitudes in the first stage, we set out to understand who DYS are from the point of view of the teachers.

We found that for both FGs participating in the research, the definition of DYS was very diverse (Table 1).

In Israel, the question of how teachers understood or defined DYS revealed that they attributed it to students who came from low socioeconomic backgrounds; to students with behavioral, emotional, or learning difficulties, learning disabilities, ADHD, LGBT minorities; and to children who had been socially excluded. In Israel, it is acceptable to characterize students with various problems, making it difficult for them to integrate optimally into school and prevent their school attendance from classifying them as at-risk or socially excluded students [37]. We found that in Israel, DYS is used as a wide "umbrella" under which we identify a multitude of problems that affect the integration and inclusion of some learners in education. It is interesting to note that most mentioned "mental problems" in their definitions, and that there was no reference to students with organic disabilities (for example, autism or physical disabilities). An explanation for this may be the existing distinction used in Israel between integration students, who are special education students who study in regular schools/classrooms, and students with various difficulties who study in regular schools.

Romanian teachers—usually those who work in big cities, who work with older students (secondary level), and who work in mainstream education—emphasize the ethical, symbolic dimension when they place a student in one category or another: students who

are marginalized, excluded students, and students whose rights are not respected. Teachers who worked in inclusive schools, and those who worked in rural areas, emphasized the socioeconomic dimension or the psycho-intellectual, ethnic dimension: students from disadvantaged families, students who do not have financial resources, those whose parents emigrated for work outside the country, or students with SEN.

**Table 1.** Cognitive component—representative statements from teachers.

| Sub-Themes | Country | Statements |
|---|---|---|
| Socioeconomic characteristics | IL | "Students who are delinquents or have links to crime families or criminals. So, these students whom I have experienced and found difficult . . . but I am referring to the extreme of the extreme"<br>"I think there is no [possibility] of preventing it [exclusion of students], the opposite, as if we constantly exclude, all the time I exclude populations. If I have an Arab student in school and we only speak Hebrew and she does not understand everything and I don't always have the time to check whether she understood—I have excluded her from what we are doing. If I have Russian students and I don't have the time to delay and see if they have understood—I have excluded them. And the more difficult language is for them, it is more difficult when we teach holidays and customs . . . when I talk to them, I put them aside all the time without realizing that I am not strengthening them, not validating who they are, where they come from and this is actually exclusion. I don't intend to, but I am at a public school . . . and this is the norm". |
| | RO | "There are many situations that can describe the vulnerability of young people: low education of parents, limited access to quality social assistance services, and after-school services."<br>"Students who are forced to work not only to support themselves but also their family and there is a risk of dropping out of school ... repeat the year" |
| Psycho-intellectual characteristics | IL | "I teach in emotional-mental classes, and it works [their integration in regular classes], but it requires a whole lot of mediation. With parents and staff and kids and constantly mediating situations. And strengthening relationships ... and forming friendships and creating a comfortable place for them in school" |
| | RO | "I work in an inclusive school—the students we have are also among those with SEN—and I think they are the ones who can be considered disadvantaged even if we don't see them that way" |
| Training | IL | "I want to say something about practical training, even if there was something that contributed to the experiences, it was not something built into teaching practice, in the training program as it was built. In other words, if there was anything I learned and benefited from experiencing such populations, it was just a coincidence. Created in the classroom I watched, and watched the teacher deal with the situation. But no—they did not direct us to such a thing. And I did not get this knowledge from school. When I came and asked and tried to understand the difficulties and dealing with students, there was even a class with emotional disorders.<br>"I could not understand. I do not know how to deal with them, with their behaviors. No, I could not." |
| | RO | "The internship I did in college helped me a lot. I went through all kinds of activities: and as a philologist at the booths, I also wanted the activity in the classroom and in the penitentiary ... and so I managed to decide what I want to do after graduating. I don't know if I still do that today."<br>"The young colleagues come totally unprepared for what awaits them: students, teachers, parents ... And they are not helped either. According to the legislation, there is a mentoring teacher who has to help his young teachers, without experience, but it is a job in which you have to do a lot of documents ... paperwork. You don't have time to teach him something practical because you have to fill in all kinds of documents." |

Source: Authors' ad hoc elaborations on research data.

In both countries, ethnicity was an important element that was used in positioning students into one category or another. In Israel, differences in the level of education were consistently found between the immigrant student population [41,42] and between students

studying in different types of education [6]. The lower level of socioeconomic development in Romania affects both the majority and ethnic minorities, although there are differences between Romanian and Roma students, which arise to a lesser extent on the basis of ethnic affiliation and are more connected to the socioeconomic status of the family.

The knowledge about DYS that was expressed by the two categories of teachers, Israelis and Romanians, also differed depending on the type of school, level of education that they taught at, the environment in which the school operated, and also depended on socio-demographic particularities of teachers (age, professional experience, etc.). Teachers who worked in schools where the school population was predominantly disadvantaged (low socioeconomic level, students with SEN, students belonging to ethnic minorities, etc.) had more complex and complete knowledge about DYS, and also had more favorable attitudes towards them. A higher level of information on what DYS meant, as well as permanent and long-term contact with DYS, had a positive impact on the types of attitudes teachers had: Teachers from both countries expressed regret for the situations from which DYS are socially and educationally excluded. Our data are in agreement with those obtained through other research, which reported that teachers who had personal interactions with populations that for various reasons were stigmatized and discriminated against (a form of disability, belonging to an ethnic minority, socio-familial, economically disadvantaged origins, etc.), generated fewer stigmatizing attitudes [43–45].

In both countries, the teachers made essential distinctions between the types of disadvantages presented by students, and their attitude was more favorable or less favorable depending on this indicator. All of the teachers from Romania and Israel who participated in the FGs argued that the degree of difficulty when working with socioeconomically disadvantaged students was lower compared to working with those who were psycho-physically disadvantaged. A study that analyzed existing reviews of inclusive education revealed that teachers have more negative attitudes towards young students with moderate learning disabilities, behavioral problems, and severe cognitive impairment, compared with children with physical disabilities and sensory impairments [46–48]. Additional studies [26] have shown that the different attitude—less favorable to students with disabilities compared to that toward socioeconomically disadvantaged students—is influenced by the fact that people with disabilities in an organizational context (school, job, community) are perceived as being dependent, with a lower level of educational or professional skills, being less productive and competitive, or as emotionally unstable [25]. Favorable attitudes come primarily from knowing and understanding more about the situation of these students. Among the individual characteristics that influence the level of knowledge of DYS, professional experience measured in years of activity is very important. The analysis of research data conducted among teachers in Israel showed that a difference in knowledge and information, rather than professional experience (teachers with experience up to 2 years, and teachers with experience over 10 years), resulted in a more favorable attitude towards DYS. In Romania, we identified the same finding: the experience and knowledge gained through working with DYS, whatever the disadvantage, led to a more favorable attitude. The effect of teaching experience on teachers' attitudes regarding inclusion is supported by previous studies [49].

The level of training is an important factor in forming attitudes [48,50,51], especially its cognitive component. One of the explanations for this correlation is that through training courses, teachers are exposed to messages/stimuli that are favorable to educational integration and inclusion; hence, they are better informed about this category of students. Moreover, through practical sessions, teachers have the opportunity to work directly with DYS, or to observe teaching activities that are carried out in schools or in classes where DYS are included. In the absence of appropriate training, teachers feel unprepared, leading to negative or neutral attitudes towards inclusion as well as hesitation to implement it [50].

In Israel, exclusion as expressed in early career teachers' answers is the product of a big system that does not have the time, skills, or space to relate to students with SEN (this does not necessarily refer to special education students). For example, for students with language

difficulties and students with behavior problems, there is an absence of knowledge and skills that teachers need. Even with adapted teaching and means of diagnosis (learning disabilities, autism, delays, etc.), teachers emphasized that they were not provided with the knowledge, skills, and tools in their academic study framework. They encountered difficulties coping with DYS during their teacher training experience framework, or were exposed to other teachers' coping methods where they did their teaching practice. In the state of Israel, the issue of including DYS has been on the educational–pedagogical agenda for a number of years. Testimonies from early career teachers informed us that in education and teaching studies and training, there is no emphasis on the DYS population unless they studied courses targeted at this population. When new teachers were asked to recommend what, in their opinion, was important to provide for teachers in training with during their studies to work with DYS for inclusion in the general education system, diverse responses were given; these ranged from theoretical knowledge about disadvantaged populations, tools, and skill to cope with students and their parents, to providing teacher-mentors, and even having teachers from the field to come in to instruct in academia. Teachers' attitude toward DYS is influenced by the feature that is considered to be the disadvantage. Thus, teachers who work with or encounter young people with SEN see rather a disadvantage in this situation, and not something that is less or not at all related to socioeconomic and material aspects. On the other hand, when it comes to educational integration, teachers who work frequently with students with SEN have a more open attitude to integration compared to teachers who only occasionally meet students with SEN. Romanian teachers should be helped by initial or in-service training to acquire not only knowledge about different categories of students, but also skills to work differently with students. Where there are cases of negative attitudes shown by teachers towards DYS, there is the opinion that this attitude originates from the lack of training that young teachers receive when they enter the system, and from a lack of support from colleagues with more experience. Some teachers believe that if current teachers in training were better prepared from a practical point of view—more class practice hours and more psychosocial training—then they would be better prepared to work with DYS. Even if new teachers enter the system less prepared in terms of training to face the reality of the system as well as encounter minimal support from older colleagues, they still have the advantage of being more open to integration and school inclusion.

### 5.2. The Affective Component

The affective component is mainly influenced by the teachers' feelings towards certain categories of students. Frustration, helplessness, and concern, alongside compassion and caring, emerged from the teachers' testimonies. Morissette and Gingras [52] considered that the affective component of attitude translates into an inner disposition of the teacher embodied in emotional reactions that are perceived/experienced whenever the teacher interacts with, or is in the presence of a student.

The testimonies of the Israeli teachers (especially the teachers with experience) revealed what they thought to be the role of the education system—it must adapt itself to the students, and not require DYS to adapt to it, in order to enable disadvantaged students to experience meaningfulness and success at school. Whom experienced teachers thought should work with DYS was the following: inclusive, accepting, and non-judgmental teachers who are capable and willing to create personal relations, and who definitely do not concentrate only on learning needs or important knowledge. Finally, what Israeli teachers believed the responsibility of policymakers and teacher training colleges should be was to produce dedicated training for all teachers, not only for those who work in SEN or DYS schools and classes.

In both countries, the relations between the favored and the DYS differed according to the institution where the teacher worked. In Romania, in schools where the population and hence the community are more homogeneous, the relations are better, and the DYS are more accepted. In schools and communities with a high degree of heterogeneity—including

students who come from socioeconomically and culturally favored backgrounds—DYS is to a greater extent marginalized. The testimonies of the Israeli participants revealed that the differences in the frameworks of SEN or those of at-risk students had more positive attitudes toward DYS than those from heterogeneous schools. According to the Romanian teachers, the attitude of rejection or marginalization towards disadvantaged young people was manifested to a greater extent by other students, classmates, or the school, and to a lesser extent on the part of teachers.

Thus, researchers [53] observed that interactions between students and teachers decreased when the concentration of poverty was higher in the school. Teachers in Romania who participated in our research confirmed this: Teachers working in rural schools had more information about their students, especially about the disadvantaged, including their families, and their educational and professional aspirations. Meanwhile, urban teachers knew less about their students; they interacted with family rarely, and only in formal environments. The results of our research revealed that some teachers cannot or do not know how to reduce the gap between DYS and those who come from socioeconomically advantaged backgrounds. In these situations, classmates and schools do not become social or emotional supports for DYS; on the contrary, they further strengthen the divide between the students. We observed that teachers did not transfer the responsibility of educational exclusion to the family, but that "the school does not make a difference" either.

In Romania, the feelings of acceptance or rejection are mainly generated by the poor socioeconomic situation of this population category, whereas in Israel, the anti-social behavior of this category of students leads to feelings of frustration or even fear (Table 2).

**Table 2.** Affective component—representative statements from teachers.

| Sub-Themes | Country | Statements |
|---|---|---|
| Feelings (towards DYS and their families; towards the education system and policy makers) | IL | "I feel that children who are less involved, children who are weak will find it difficult to push themselves and if parents do not help, it is even more difficult and also much more difficult to open one's heart and establish contact when communicating with parents is not good generally, to create something to support a child, to create some sort of helping front. So, if parents also cannot support [a child] themselves and it is also hard for them to establish good contact with a school, I think it will be very difficult to hold onto a child as if to lift such a child or support him" |
| | | "I find it difficult to work with the populations I mentioned [at-risk students, behavior problems, social and geographical peripheries], I admit it and I avoid it . . . I will not go to schools in such areas" |
| | | "They (DYS) are delinquent, theyare a delinquent population, they have delinquent speech, delinquent behavior, it's difficult for me... it scares me... I'm afraid..."<br>"I don't know how it is in regular classes, but I know that in special education classes they really struggle to accept every child because they are children who have been 'ejected' from all sorts of systems, from all sorts of frameworks and if they are not at our school, then they are simply on the streets, so they try very, very hard to accept these children and meet their needs and fight for every child. And also, children, simply had their needs met during the Corona period, those they knew were alone at home, who had no support and would not learn, they were allocated an assistant to be with them and help them. Although they are not considered education or, really everything they could. It was very important for them" |
| | RO | "Sometimes parents compensate for the shortcomings they have faced themselves, they give their children a lot ... they exaggerate ... They send them to school with a lot of money, they dress them according to the latest fashion ... but not all children have these possibilities and those who have humiliated those who do not. This is about the education that children receive in the family." |
| Communication skills | IL | "They find friends similar to them and then they become a small group" |
| | RO | "The most difficult thing is to work and communicate with young people who have psycho-intellectual problems and their parents do not accept this. You as a teacher cannot help him if the parent does not recognize: you cannot make a curriculum adapted to the newcomers, you cannot evaluate him so as to respect his requirements ... And the hardest thing is for the parent to recognize the fact that he has a child with special needs" |

Source: Authors' ad hoc elaborations on research data.

In Romania, the relationship between teachers and parents of DYS is closer in rural areas than in urban areas. Moreover, Romanian researchers [54] demonstrated that during the communist period as well as today, "spontaneous integration" is practiced in rural areas because of the closer relations between school and family: people show understanding and acceptance towards all members of the community because it is evaluated on the basis of other criteria unrelated to the amount of financial resources, the intellectual level, the professional position, etc. In urban schools, those rated as schools of excellence, the proportion of DYS is much smaller, and where this category of student exists, the relationships are more distant both between students and between students and teachers.

*5.3. The Behavioral Component*

Through the analysis of the behavioral component of the attitude, we aimed to understand what actions and concrete activities teachers performed that were in favor of DYS. The behavioral component can also be seen in the fact that a teacher can adopt techniques and learning methods that support disadvantaged students, maintain direct contact with the students 'families or, on the contrary, avoid such behaviors and thus discourage students' aspirations, and increase the distance between school and these categories of students.

Through the analysis of the behavioral component of the attitude, we aimed to understand what actions and concrete activities teachers performed that were in favor of DYS (Table 3). More precisely, we were interested in understanding if working with DYS involved the application of different pedagogical methods and techniques compared to other categories of students, and if the teachers communicated with each other and with specialists in order to identify solutions that would support disadvantaged students.

In addition to the transmission of knowledge, teachers must also perform other tasks: managing the class of students, which means enforcing some rules, norms, and values; and managing commitment, which involves the formation of students' motivation to learn [55]. The analysis of the cognitive component showed us that in terms of teacher training for this responsibility, both countries showed deficiencies. As a result, in their statements teachers used words and expressions such as, *"I tried to involve," "very difficult,"* or *"I do not know how to deal with them"* (Table 3). The moment when the teachers claimed, as we observed in the statements, that was is difficult to work with DYS, they felt a need for support from the family and from other specialists; this meant that teachers faced difficulties with class management in terms of managing the students, and managing their commitment.

In both countries, results obtained from other researchers were also confirmed [39]: Teachers who are in favor of general access to education tend to diminish the importance of sociocultural factors on the pedagogy to support DYS. Of course, this effort cannot be supported only by teachers, but requires the support of authorities and families.

From the analysis of the behavioral component, we observed that in both education systems, professional experience, measured in years of activity, as well as the type of school in which they worked, had a positive effect on teachers' attitudes regarding educational inclusion being more important than training in relation to DYS. This result was consistent with previous research [49], which reported that teachers with more teaching experience tended to believe more that inclusive education leads to positive changes among disadvantaged students as well as among those who are not.

In Israel, early career teachers were very surprised by the lack of tailored training for working with DYS during their studies, except for those whose graduate degrees were in special education and were specialized in disabilities encountered in students. When inexperienced teachers were in a situation of working with DYS, they adapted the methods, but pointed out that they were not provided with the knowledge, skills, and tools during their academic training. Contact with other colleagues, as well as practical training with DYS helped them to improve their effectiveness in the classroom.

In Romania, there is no differentiated training for teachers; those who will work with DYS should seek additional training so that they can acquire specific work methods and techniques. Each teacher must face it alone, or with the support of colleagues who

have professional experience, to find the best pedagogical methods and techniques to work with DYS. Studies that examined the effect of preparation courses for working with disadvantaged students [56] show that formal teaching with structured experience in field work may foster positive changes in attitudes towards inclusive education.

**Table 3.** Behavioral component—representative statements from teachers.

| Sub-Themes | Country | Statements |
|---|---|---|
| Classroom management, commitment management | IL | "I went in and tried to conduct a lesson and didn't succeed, no way. Later I sort of developed methods, but it is not something I achieve through education [studies]" |
| | RO | "No one teaches you how to organize yourself, how to interact with these students. You learn on the go, you adapt according to the situation. Talk more with your colleagues." |
| Pedagogical methods and techniques | IL | "I want to say something about practical training, even if there was something that contributed to the experiences, it was not something built into teaching practice, in the training program as it was built. In other words, if there was anything I learned and benefited from experiencing such populations, it was just a coincidence. Created in the classroom I watched, and watched the teacher deal with the situation. But no—they did not direct us to such a thing. And I did not get this knowledge from school. When I came and asked and tried to understand the difficulties and dealing with students, there was even a class with emotional disorders. "I could not understand. I do not know how to deal with them, with their behaviors. No, I could not." |
| | RO | "It is very important to involve all students in the activity. I witnessed classes in which the teachers did not work with some students. They were there, he had forgotten them...they were given something to draw or copy a text...And I asked: can this student do that or do they not want to work with him?" |
| Collaboration between teachers, between teachers and DYS, and with other categories of specialists (psychologists, school counselors, speech therapists, etc.) | IL | "I remember when I was doing teaching practice during my studies at such a last chance school, where they did amazing work, amazing with the children—at-risk children . . . and I remember how impressed I was with the whole process there. And I said to myself, this would never happen in a regular school as if no framework could give them what they received at that place" "I want to admit and say that I do, I do give up on children whose parents ah, whom I tried to involve and they were not, they were not involved." |
| | RO | "You know the situation in Romania: school counselors and psychologists are very few and they run from one school to another... they don't have time to get to know the student, to help the teachers, to establish an appropriate curriculum together with him, to talk with the parents.. very difficult. Each teacher copes as he can and as he knows best." |

Source: Authors' ad hoc elaborations on research data.

Furthermore, parental cooperation is a critical component in working with disadvantaged students. Contact with parents serves as a type of power multiplier for teachers who cope with the complexities of educational work and teaching. When parents do not cooperate, teachers may despair and give up on a student. It was observed that teachers from Israel try to cope alone with individual students and their difficulties; if unsuccessful, they turn to help within the school, for example, from school counselors. If problems persist, they turn to parents for help.

In Romania, the biggest problem of cooperation with the family derives from the fact that many parents either do not acknowledge, or find it very difficult to admit, that they have a child with problems, whatever their nature is. The negative attitude manifested by the teachers towards DYS was largely due to the insufficient support received.

When the teachers claimed that, "the state does not do enough. Not only doesn't it do enough in my view and that of my friends in the staffroom, but we are also very well aware of what happens in the field," that means support in training, as well as support in relation to other specialists, but also in terms of legislation; it is obvious that dissatisfaction turns into a negative attitude towards DYS.

The hope for this category of the school population, in both countries, is represented by the fact that, "even if it is not specific to all teaching staff and all schools, it is noted that

many gifted and talented persons with a range of options choose the teaching profession because of a sense of mission." [14] (p.40).

## 6. Limitations

One of the limitations of the current analysis stems from the fact that we only used the data collected through FGs with teachers, and not from other categories of data (students, specialists who collaborate with schools and work with DYS), reflecting only one perspective on analyzed topics. The second limitation has to do with the characteristics of a qualitative research: a small sample size, and a lack of representativeness of the sample, since the selection was not made by objective, probabilistic methods. However, the participants were selected as a purposeful sample considering the maximum variation of participants sought (years of study, age, and gender).

The collection of data through online platforms is susceptible to limited research. Direct contact in the same room of the moderator with the participants of the FG as well as the physical presence of all the participants, facilitates communication and the exchange of ideas, since it allows the establishment of direct, closer contact.

Another limitation that likely influenced the research data is the fact that during its development, the entire population was facing the effects of the COVID-19 pandemic, which diminished the importance of other problems, and changed attitudes towards other social issues, including those of DYS. The analysis of other data we collected, as well as our intention to complete this study with a quantitative one based on a questionnaire, will allow us to obtain better results that reflect, as objectively as possible, attitudes towards DYS in the two countries included in the analysis.

## 7. Conclusions and Implications for Practice

The analysis of the research results revealed the main factors likely to influence teachers' attitudes towards DYS. It is often the case that ethnic origins and socioeconomic status influence teachers' attitudes toward students, in such ways that teachers tend to categorize them as DYS. The environment (settings where teachers work) impacted teachers' attitudes toward their DYS; those who worked in normative and heterogeneous settings were less open and accepting than those from SEN schools and classes, or those who were from settings for at-risk students in Israel or rural schools in Romania. The socio-demographic characteristics of the teachers (age, seniority, experience) influenced the their attitudes towards DYS. In both countries, the most negative attitudes were towards delinquent and disabled students. Teachers without special training to work with DYS (and who therefore, lacked relevant skills) in many cases had more negative perceptions compared to teachers who received such training. Teachers found it difficult, at best, and did not know, at worst, how to prevent social exclusion. In addition, teachers' attitudes were affected by their emotional experience; meetings with DYS often evoked feelings of helplessness, frustration, and anger at superiors, authorities, or parents. Acquaintance and cooperation with colleagues, relevant professional parties, and parents influence teachers' attitudes towards DYS, as well as their perceptions of their abilities to succeed in their work toward educational inclusion.

The context in which education for all was implemented in Israel and in Romania are quite different, although some differences and similarities in their profiles of attitudes towards DYS can be clearly identified. The history of each country's commitment to ensuring access and success in education for all categories of students, and the associated historical legacies of diversity in society in general (education in particular), clearly mediated the nature and quality of educational development in both countries. The present study revealed a series of important issues for educational policy and practice in Israel and in Romania regarding DYS. First of all, in both countries, we observed that teacher training is important, and that teachers' knowledge must be constantly updated with the findings of various research, testing, and implementation of pedagogical methods regarding DYS. Moreover, authorities in both countries must take into account the professional experience

of teachers who successfully work with DYS, because they represent "resource persons" who can contribute to promoting quality education for all; they also facilitate access to more sustained support for teachers who work with disadvantaged school populations. At the same time, the two education systems must prepare new generations of teachers, pay more attention to their training, and rely less on on-the-go learning. Another direction that deserves further exploration is the development of cross-national analyses through expanding and diversifying the countries that are compared. This analysis of qualitative data helped us to identify the most frequent and important themes and sub-themes regarding attitudes of teachers towards DYS. Along with detailed analyses from other FGs (with students, future professionals, and specialists from other sectors of activity who collaborate with educational institutions), they represent a starting point to carry out quantitative research on a representative sample. This initial analysis showed us that despite differences related to the economic development, culture, and history of each country, certain aspects of ensuring a quality education for all are identical.

**Author Contributions:** Conceptualization, P.L.-S. and G.N.; methodology, P.L.-S. and G.N.; validation, P.L.-S. and G.N.; formal analysis, P.L.-S. and G.N.; investigation, P.L.-S. and G.N.; resources, P.L.-S. and G.N.; data curation, P.L.-S. and G.N.; writing—original draft preparation, G.N.; writing—review and editing, G.N.; project administration, P.L.-S.; funding acquisition, P.L.-S. All authors have read and agreed to the published version of the manuscript.

**Funding:** This research was supported by "ORANIM" Academic College of Education.

**Institutional Review Board Statement:** The study was conducted in accordance with the Declaration of Helsinki, and was approved by the Institutional Ethics Committee of Oranim Academic College of Education (Date: 15 March 2021, Confirmation number: 111).

**Informed Consent Statement:** All individuals included in this section have consented to the acknowledgement.

**Acknowledgments:** This article is based upon work from the Faculty of Social Sciences and Humanities, Oranim Academic College of Education Israel in the project: The attitude of teachers towards disadvantaged young students. Israel-Romania comparative analysis.

**Conflicts of Interest:** The authors declare no conflict of interest.

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
