# Peer review of "The Attitudes of Teachers towards Disadvantaged Young Students: Israel–Romania Comparative Analysis"

_sustainability, doi:10.3390/su141912468_

Round 1

Reviewer 1 Report

The study addresses an important issue, but some methodological corrections are needed.

In my opinion, it was not appropriate to give a theoretical framework in the method section. The theoretical framework can be combined and presented with contextualization.

The purpose of the research should be expressed in a paragraph or 2-3 sentences. Research questions should be clearly stated.

The method section should be developed. The model of the research should be presented. The sample should be handled as a separate topic. It should be stated how the data collection tool was developed. It should be explained how the validity and reliability are ensured both in the development of the data tool and in the data analysis.

In the analysis part, the views of the participants were included and the researcher interpreted these views. The themes reached while presenting the findings should be presented. The reader should be able to see which concepts/themes have been reached under each title or dimension while reading the findings. To summarize, the themes reached as a result of the content analysis should be included in the tables.

Findings are presented in the Discussion section. It is necessary to present the findings under the Results title. In the discussion, the previous related studies should be given more place and the results should be discussed. It is very important to increase the number of studies in the discussion section. In addition, current studies should be included.

The conclusions section should also include limitations and recommendations.

Author Response

Dear reviewer,

Thank you for your interest in our article and especially for the important suggestions you sent us.

In the revision of the manuscript, I created a special chapter for the theoretical framework to explain more clearly the perspectives from which we approach the problem of disadvantaged young people from the two countries included in the comparison: Romania and Israel.

We have reformulated the research question to be clearer for us and for the readers of the journal.

This section dedicated to the methodological framework has been greatly improved following your suggestions:

  • We argued why qualitative research based on focus groups is much more appropriate for this topic compared to other qualitative methods (interview, for example);
  • We explained in detail how the focus group participants have selected: the selection method, the applied criteria, the method of data collection, the stages of the focus group organization, and the characteristics of the participants from the two countries;
  • We explained in detail how the data collection was carried out: the factors we took into account in this process, the way we managed the methodological risks and limits so as to ensure the validity and reliability of the research;
  • We detailed the way in which I analyzed the data: the system of analysis categories, the ordering of the collected data based on frequencies, and the rules that I applied in ordering the content.

We organized the tables in the article and especially their contents. Also, in the results and discussions chapter, I have better emphasized each theme and subtheme. We have reorganized the content of the article to better emphasize the results and the discussions about them. We have also added a separate chapter regarding the limits of our research. Another important aspect is that by improving the article we have expanded the number of studies we refer to, and we have better connected the results obtained by us with those of other researchers.

Thank you very much for your patience and effort and we hope to have a much better article at this moment.

Best regards, 

Gabriela Neagu

Pazit Levi -Sudai

Reviewer 2 Report

I have read the article with pleasure. I find the topic important and timely, especially in the light of comparison between such different countries and their educational systems. 

My objections refer to three aspects: the way of expressing things, the way of presenting the study, and the references. 

As to the first aspect, the language is at times too obscure and it's difficult for the reader to work out what the Authors communicate: p. 4 (158-162 and others): Is it a definition or the authors' interpretation? Or lines 204-210, or the last paragraph in 3.1.3? The sentences are sometimes too long, which makes an impression of verbosity, e.g. a sentence in 99-100 - What new does it contribute? Hardly anything. The same goes with inclusions in brackets (l. 98) - the choice of the tense makes its relevance in the present time, or unnecessary words (and in l. 27, people in l. 34, a in l. 395, lack of congruence (cognitively but affective, why not affectively in l. 152, GLBT - why not LGBT in l. 348). This all implies that the linguistic correction could be useful. 

As to the presentation of the study, it could be more transparent if the authors presented the procedure in a tabular form, as the study involved a few stages. The same recommendation refers to the participants of the FG in the two countries - how many were there, and other demograhic information (seniority, level of education, etc. could be presented in a table. 

The references are on the whole sufficient but, if possible, more recent references to publications after 2017 could be added. 

I have no reservations to the analysis and teh conclusions drawn from the data. 

Author Response

Dear reviewer,

Thank you for your attention to our article and for the suggestions made. All of them are of real help to us.

Regarding your first suggestion - the language - the most important measure I have taken is to call for a professional translation. In this sense, I contacted the representatives of the journal to choose the best solution together.

We have significantly modified the structure and content of the article as follows:

  • We argued why qualitative research based on focus groups is much more appropriate for this topic compared to other qualitative methods (interview, for example);
  • We explained in detail how the focus group participants have selected: the selection method, the applied criteria, the method of data collection, the stages of the focus group organization, and the characteristics of the participants from the two countries;
  • We explained in detail how the data collection was carried out: the factors we took into account in this process, the way we managed the methodological risks and limits so as to ensure the validity and reliability of the research;
  • We detailed the way in which I analyzed the data: the system of analysis categories, the ordering of the collected data based on frequencies, and the rules that I applied in ordering the content.

The improvement of the article mainly as a result of the suggestions of the reviews helped us a lot to correct and improve the references with the recent literature.

Thank you once again.

Best regards, 

Gabriela Neagu

Pazit Levi-Sudai

Reviewer 3 Report

Dear authors:

Your study is very relevant.

It seems to me that the flaws are due to the fact that only part of the research is in this article, which made it difficult to write. So, here are my suggestions to improve your manuscript:

-do not use the word "etc" (for example: line 21, line 273); 

- the statement is too generic: "Our study belongs to the category of those that provide important information for the policies and practice of quality education for all.” (line 20-21);

- The creation of subsection "3.1. Theoretical Background" does not seem appropriate. 

I suggest the following change in the organization of section 3:

3.  Materials and Methods

3.1 Methodological framework

The concept around which we built research is that of attitude. The next three pillars of research are inclusive education, educational integration, and disadvantaged young people (DYS). The following review addresses the basic concepts of this research.

3 2.Procedure 

3.3 Data analysis 

- You should rethink the title of sub-section 3.1.1 and figure 1: 'Attitudinal dimensions'. It is not about dimensions. This should also be reviewed in section 4;

- “3.2.3. Results " needs to be rewritten. All described in this section could be included in the "Procedure" section;

-  You have to explain this: "The research included the participation of 10 teachers from Romania and 16 teachers 302 from Israel " (lines 302-303). Why is the number different? 

- Clarify in Subsection 3.2.2:

"people's attitudes (what they feel)"- It is not takes account the figure 1. This needs to be reviewed.

It seems to me that (see Figure 1) "people's attitudes" flows from:  affective dimension +cognitive dimension+ behavioral dimension.

-I don't understand what they wrote in lines 280-282 ("competencies" and "actions"????);

- As you have described section 3.2.2, there is no way to understand section 4. I suggest you be more precise in the description of the "data analysis" by dimension, as there must be a relationship between the tables in section 4 and what you describe in section 3.2.2. It does not seem appropriate to write "4. Discussions". I suggest, for example: "4. Results and Discussion" or "4. Discussion" and a section with results;

- Section 5: You should write a paragraph with the results of the study, as you wrote in the abstract;

- Only 4 references are from last 5 years. I suggest you revise that aspect.

Rew

Author Response

Dear reviewer,

Thank you for the interest shown in our article, for the appreciation made, and especially for the very important suggestions.

Your suggestions helped us to improve our article.

We observed, like yours, that there were indeed many such expressions and words -” etc.”-  or sentences with a very general character. We corrected the article and removed and/or improved the way of expression.

We reorganized the entire article and created new chapters and sub-chapters that better reflect the theoretical framework, the methodological framework, the results and discussions, the limits of the research as well as the conclusions and practical implications.

We have created a detailed chapter on the methodology used as follows:

  • We argued why qualitative research based on focus groups is much more appropriate for this topic compared to other qualitative methods (interview, for example);
  • We explained in detail how the focus group participants have selected: the selection method, the applied criteria, the method of data collection, the stages of the focus group organization, and the characteristics of the participants from the two countries;
  • We explained in detail how the data collection was carried out: the factors we took into account in this process, the way we managed the methodological risks and limits so as to ensure the validity and reliability of the research;
  • We detailed the way in which I analyzed the data: the system of analysis categories, the ordering of the collected data based on frequencies, and the rules that I applied in ordering the content.
  • We explained and argued the reason why there are differences between the number of participants in the focus group - mainly because the two countries but also their education systems are different in terms of multiculturalism (which is also reflected at the level of the system of education from each country) and we considered this to be very important to be represented in our research by the participants of the FG.

We followed your suggestions and redid the discussion chapter, renaming it "results and discussions".

We have renamed the sub-themes analyzed and presented in the tables to be clearer for the readers of the journal.

As a result of the suggestions of the reviews, we have improved and expanded the content of the article and at the same time, we have included new, very current references in the text.

Thank you for your support.

Best regards, 

Gabriela Neagu

Pazit Levi-Sudai

Round 2

Reviewer 3 Report

Dear authors:

 I appreciated the improvements made to the manuscript. It has more soundness.

Rew